# Comprehensive characterization of the bacterial community structure and metabolite composition of food waste fermentation products via microbiome and metabolome analyses

Hongmei Li[1,2], Xiaoyang Lin[1], Lujun Yu[1], Jianjun Li[1], Zongyu Miao[1], Yuanzheng Wei[1], Jin Zeng[1], Qi Zhang[3], Yongxue Sun[2]*, Ren Huang[1]*

1 Guangdong Provincial Key Laboratory of Laboratory Animals, Guangdong Laboratory Animals Monitoring Institute, Guangzhou, China, 2 College of Veterinary Medicine, South China Agricultural University, Guangzhou, China, 3 Shenzhen Teng Lang Renewable Resource Development Co., Ltd, Shenzhen, China

* 1649405216@qq.com (RH); sunyx@scau.edu.cn (YS)

**Data Availability Statement:** The original data of 16S rRNA gene sequencing was submitted to the

## Abstract

Few studies have characterized the microbial community and metabolite profile of solid food waste fermented products from centralized treatment facilities, which could potentially be processed into safe animal feeds. In this study, 16S rRNA gene sequencing and liquid/gas chromatography-mass spectrometry were conducted to investigate the bacterial community structure and metabolite profile of food waste samples inoculated with or without 0.18% of a commercial bacterial agent consisting of multiple unknown strains and 2% of a laboratory-made bacterial agent consisting of *Enterococcus faecalis*, *Bacillus subtilis* and *Candida utilis*. Our findings indicated that microbial inoculation increased the crude protein content of food waste while reducing the pH value, increasing lactic acid production, and enhancing aerobic stability. Microbial inoculation affected the community richness, community diversity, and the microbiota structure (the genera with abundances above 1.5% in the fermentation products included *Lactobacillus* (82.28%) and *Leuconostoc* (1.88%) in the uninoculated group, *Lactobacillus* (91.85%) and *Acetobacter* (2.01%) in the group inoculated with commercial bacterial agents, and *Lactobacillus* (37.11%) and *Enterococcus* (53.81%) in the group inoculated with homemade laboratory agents). Microbial inoculation reduced the abundance of potentially pathogenic bacteria. In the metabolome, a total of 929 substances were detected, 853 by LC-MS and 76 by GC-MS. Our results indicated that inoculation increased the abundance of many beneficial metabolites and aroma-conferring substances but also increased the abundance of undesirable odors and some harmful compounds such as phenol. Correlation analyses suggested that *Leuconostoc*, *Lactococcus*, and *Weissella* would be promising candidates to improve the quality of fermentation products. Taken together, these results indicated that inoculation could improve food waste quality to some extent; however, additional studies are required to optimize the selection of inoculation agents.

serial access archive (SRA) under the registration number PRJNA751165.

**Funding:** The author, HML, was supported by the Guangzhou Basic and Applied Basic Research Project [Grant No. 202102020233]. The URL of funder website: http://kjj.gz.gov.cn/gsxx/content/post_7285885.html The funder had no role in study design, data collection and analysis, decision to publish, or preparation of the manuscript.

**Competing interests:** NO authors have competing interests Enter: The authors have declared that no competing interests exist.

## 1. Introduction

According to the 2021 Food Waste Index report released by the United Nations Environment Program, an estimated 931 million tons of worldwide food waste are generated each year [1]. Currently, more than 90% of food waste in developing countries is still mixed with municipal solid waste and is either sent to landfills or incinerated [2]. Food waste is derived from human food and is therefore highly nutritious and has a similar nutritional composition to that of animal feed. The conversion of food waste into animal feed has environmental benefits, in addition to being low cost and providing added value. Therefore, the use of food waste as animal feed has garnered increasing attention among environmental researchers [3–5]. A few regions in the world have begun to use processed food waste as animal feed in modern pig, chicken, and fish farming systems [6–8].

China is among the countries that produce the most food waste worldwide [1]. Food stalls, restaurants and canteens in China generate approximately 45 million tons of food waste each year [9]. Since 2010, China has been implementing an exploration program to convert food waste into animal feed, and has established demonstration projects in many cities. Three typical treatment processes (i.e., heat treatment, fermentation, and coupled hydrothermal treatment and fermentation) are usually used in centralized food waste treatment centers. Food waste processed using either of the aforementioned procedures is considered to have some nutritional value and meets relevant microbiological and chemical contaminant standards, making food waste a promising alternative to be used in animal diets [10]. The processing of food waste via fermentation is generally believed to improve the nutritional value, increase beneficial bacteria and digestive enzymes, improve the palatability of raw materials, and prolong shelf life [11–14]. However, most assessments of microorganisms in food waste fermented feeds have focused only on a few probiotics and pathogens. Similarly, most studies on composition of food waste fermented feeds have mostly focused on the determination of major nutrients. Few studies have comprehensively investigated the microbial community and metabolite profiles in fermented feed, as well as the fermentation products of food waste in centralized treatment centers. These unknown microorganisms and metabolites may have a significant impact on the stability and quality of fermented food waste and animal intestinal health [15].

In this study, heat-treated food waste obtained from a centralized treatment facility was used as the substrate, and corn flour and soybean meal powder were used as auxiliary materials, as these materials can be used to adjust the moisture content and provide inter-particle gaps. Further, this study employed two kinds of compound bacterial agents that can significantly improve the nutrient profile of the food waste fermentation products, reduce odors, and lower the pH. One of them is a widely used commercial inoculant, whereas the other is a laboratory-made compound inoculant/starter. Solid-state fermentation was conducted, and the changes in the composition of the microbial population and the changes in the product composition during the fermentation of food waste were discussed. An Illumina HiSeq 2500 sequencer was used to characterize the bacterial 16S rRNA gene in the fermentation products, and the metabolite compositions of the products were analyzed via liquid phase mass spectrometry (LC-MS) and gas mass spectrometry (GC-MS). This study thus clarified the changes in microbial and chemical composition during fermentation by correlating the succession of microbial communities and metabolites present in the fermentation product. Therefore, our findings provide a basis for the evaluation of the nutritional value and fermentation quality of food waste-derived animal feeds.

## 2. Materials and methods

### 2.1 Raw materials and fermentation procedures

The food waste was collected at a demonstration project (Shenzhen Teng Lang Renewable Resource Development Co., Ltd), which collects and processes 200 tons of food waste per day.

The food waste was then processed by mixing, crushing, and removing non-food impurities, followed by hydrothermal treatment, dehydration, and degreasing according to industry standards. Heat-treated food waste samples were aseptically collected from December 2019 to December 2020 for general analyses. Afterward, 2.0 kg of heat-treated food waste were randomly collected each day for a full week and stored at 4˚C until the sample collection was completed. Next, 8% cornmeal and 5% soybean meal powder were added to the food waste to obtain a food waste medium with a moisture content of approximately 65%. According to its manufacturers, microbial inoculum #1 contains Pediococcus lactis and Saccharomyces cerevisiae, among others, and was purchased from Yi chun Qiang sheng Biotechnology Co., Ltd. Microbial inoculum #2, on the other hand, is a probiotic culture preserved in the laboratory that contains *Enterococcus faecalis* (E), *Bacillus subtilis* (B), and *Candida utilis* (C). The concentrations of these three strains were 108 cfu/mL in culture and the culture was prepared at an E: B: C ratio of 3: 1: 2. Next, three groups of heat-treated food waste with added corn and soybean meal were sampled and inoculated with 0.18% commercial inoculum 1 (T1) and 2% laboratory-made inoculum 2 (T2) according to the product instructions, whereas the third group was left without inoculation (CT) to serve as a control. Three parallel samples were obtained from each group, mixed thoroughly, and transferred to a fermentation bag equipped with a one-way breather valve (Ruduoduo Biotechnology Co., Ltd., Beijing), then incubated for 4 days at 28˚C according to the fermentation conditions explored in the previous stage (S1 Fig). During the fermentation process, the materials in the fermentation bag were kneaded and stirred every 6 hours until the fermentation process was complete. The products were collected and stored at -80˚C for subsequent bacterial composition and metabolite profiling analyses.

## 2.2 Detection and characterization of fermentation products

The pH value of the heat-treated food waste and fermentation products was measured with an S8 pH meter (Shanghai Mettler-Toledo Instrument Co., Ltd.). After drying at 65˚C for 48 hours, nutritional indicators including crude protein (CP), crude fat (EE), crude fiber (CF) and ash (Ash) were characterized using the proximate analysis method [16, 17]. Reducing sugar content were determined via the titration method as described by Jiang et al. [18]. *E. coli* was detected using *E. coli* chromogenic medium (Guangdong Huankai Microbial Technology Co., Ltd., China). *Lactobacillus* spp. (LAB) and mold were quantified using MRS medium and Bengal red agar medium, respectively. *Salmonella*, *Staphylococcus aureus*, aflatoxin, and vomitomycin were tested according to the recommended methods in the "Feed Hygienic Standards" (GB13078-2001) [19]. Organic acids were characterized via HPLC according to the conditions and procedures described by Nisperos-Carriedo et al. [20]. Aerobic stability was assessed as described by Acosta et al. [21]; the food waste fermentation products were placed in an insulated polyethylene box with an open lid at 21˚C and covered with gauze. Aerobic deterioration was defined and reported as the number of hours at which the temperature rose 2˚C above the ambient temperature.

## 2.3 DNA extraction and 16S rRNA gene sequencing

The genomic DNA of the food waste fermentation product was extracted using the HiPure Stool DNA kit (model D3141, Magen Biotechnology Co., Ltd, China) according to the manufacturer's instructions. DNA integrity was then inspected via 1% agarose gel electrophoresis. A NanoDrop spectrophotometer (model NanoDrop 2000, Thermo Fisher Scientific, USA) was used to determine the DNA quality [22]. Barcode-specific primers 341F (`CCT ACG GGN GGC WGC AG`) and 806R (`GGA CTA CHV GGG TAT CTA AT`) were used to amplify the

16S rDNA V3-V4 hypervariable region [23]. The PCR product was purified using AMPure XP Beads and quantified with a Qubit 3.0 fluorometer after purification. Sequencing was conducted in the Illumina Hiseq2500 platform with the PE250 strategy at Gene Denovo, Guangzhou, China. The original data was submitted to the serial access archive (SRA) under the registration number PRJNA751165.

The original Illumina FASTQ data were processed using QIIME software version 1.9.1. After removing all the chimeric tags and quality control, a read trimming tool (Trimmomatic) was used to obtain high-quality sequences for downstream analysis. UPARSE9 (version 9.0) was used to perform operational taxonomic unit (OTU) clustering analysis on the obtained sequences based on a 97% similarity threshold, after which taxonomic analysis was conducted through the naive Bayes model of the RDP classifier of the SILVA (version v123) database. Alpha diversity indices such as Sobs, Ace, Chao1, and Shannon were calculated in QIIME, after which visualizations of the OTU thinness curve and rank abundance curve were created [24]. Principal coordinate analysis (PCoA) was conducted using unweighted_unifrac distance and non-metric multi-dimensional scaling (NMDS) based on bray distance. The relative abundance of OTUs at the phylum, class, order, genus, and species level of all samples were compared. Furthermore, PICRUSt2 was used to infer the KEGG pathways of each of the elucidated OTUs [25].

## 2.4 Liquid chromatography-mass spectrometry analysis

To conduct metabolomic analyses, the fermentation products of food waste were first pre-treated with methanol. A Waters LC-MS system (Waters, UPLC; Thermo, Q Exactive) and Acquity UPLC HSS T3 columns (2.1 × 100 mm 1.8 μm) (Waters, Milford, MA, USA) were used for separation. The Compound Discoverer software (Thermo company) was used to extract and preprocess the LC/MS detection data and normalization. The results were exported as a matrix containing information such as retention time (RT, Retention time), molecular weight (CompMW), observation volume (sample name), number of extractable substances (ID), and peak intensity. For quality assessment, the online human metabolite database (http://www.hmdb.ca/) was used to identify the detected metabolites. Six replicates were analyzed for each group for detection and analysis.

## 2.5 Gas chromatography-mass spectrometry analysis

To compare the changes in the volatile components of the samples, GC-MS was employed to identify the composition of the gas released from the fermentation samples. To achieve this, 2.5 g of food waste fermentation samples were transferred to a 20 ml headspace bottle. After 3 hours, a 57310-U solid-phase microextraction needle (Sigma-Aldrich, USA) was used to collect volatile substances in the top space of the bottle for 30 minutes. A Thermo Fisher Trace 1300 instrument (USA) equipped with a DB-WAX column (30 m × 0.25 mm × 0.25 μm) was used for GC-MS analysis [26]. Helium was used as a carrier gas (99.999%) at a 1.0 mL/min constant flow rate. The GC analysis conditions included an initial temperature of 60°C for 3 minutes, which increased first to 145°C at a 6°C/min rate, then increased to 240°C at 15°C/min and maintained at this temperature for 3 minutes. MS was conducted with electron ionization, an interface temperature of 230°C, a 70 eV electron impact, and a mass range of 35~550 m/z in full scan mode. The NIST library was used as a mass spectrum search library. Six replicates were analyzed for each group for detection and analysis.

## 2.6 Data statistics

The effect of bacterial inoculation on food waste fermentation properties and bacterial community structure was assessed via one-way analysis of variance (ANOVA) using the SAS 9.3

software (SAS Institute Inc., Cary, NC, USA). P-values <0.05 were deemed statistically significant. The metabolite data were analyzed by partial least squares discriminant analysis (PLS-DA) using the SIMCA-P software. PLS-DA is determined by the goodness of fit parameter (R2X) and the predictive ability parameter (Q2). Differentially expressed metabolites were identified using the PLS-DA model coupled with the first principal component of the VIP (variable importance in the projection) value (VIP> 1), as well as Student's T-test (P <0.05). GraphPad Prism software version 6.00 (GraphPad Software, San Diego, California, USA) was used to calculate the Spearman correlation coefficient to evaluate the relationship between differentially expressed metabolites and bacterial classification at the genus level. Heat map were generated using TBtools [27]. Strain and metabolite correlation maps were generated using Cytoscape [28].

# 3. Results

## 3.1 Heat-treated food waste materials contained adequate nutrient levels and low concentrations of harmful substances

This study adopted on-site random sampling and testing and reported the composition of a total of 13 heat-treated food waste samples from December 2019 to December 2020. The data composition is the original data ± SD, as shown in Table 1. The heat-treated food exhibited a high crude protein content (34.9%, DM). Further, the high sugar reduction (5.54%, DM) in the samples indicated that the heat-treated food waste contained substrates that promoted bacterial fermentation. The low coliform bacteria count in the samples were likely attributable to heat treatment. Other harmful substances were undetectable or were present in negligible concentrations. Additionally, the analysis of the differences between batches indicated that the ingredients and proportions of each month were relatively consistent.

## 3.2 Inoculation with different bacterial agents affected the smell, appearance and physicochemical properties of the fermentation products

The heat-treated food waste products were randomly sampled, grouped, and fermented according to the methods described above. After 96 h of fermentation, the three treatment groups exhibited strong changes in smell and color compared with the raw heat-treated food waste materials. Further, as demonstrated in Table 2, both the T1 and T2 inoculants promoted crude protein accumulation, particularly in the T1 group (P<0.05). The pH values of the T1 and T2 groups were lower (P <0.05). At the same time, the aerobic stability of T1 and T2 was

**Table 1. The characteristics of the raw materials.**

| Index | Value | Index | Value |
|---|---|---|---|
| Crude protein (%, DM[a]) | 34.9±2.4 | Enterobacteria (CFU /g FM[b]) | 137±56 |
| Ether extract (%, DM[a]) | 9.1±1.2 | Molds (CFU /g FM[b]) | 36±11 |
| Crude fiber (%, DM[a]) | 8.2±1.0 | Salmonella (CFU /g FM[b]) | ND[c] |
| Reducing sugar (%, DM[a]) | 5.5±2.1 | Staphylococcus Aureus (CFU /g FM[b]) | ND[c] |
| Ash rate (%) | 6.9±0.3 | Aflatoxin B1 (μg/kg) | UMD[d] |
| pH | 4.52±0.41 | Deoxynivalenol (μg/kg) | UMD[d] |

Note: the data are shown as means ± standard deviation.

a Dry material.

b Fresh material.

c Not detected.

d Under the detection limit of the analysis method.

**Table 2. Fermentation characteristics of food waste fermentation products.**

| Item | Treatment | | |
|---|---|---|---|
| | Control | T1 | T2 |
| pH | $3.92\pm0.09^a$ | $3.64\pm0.02^c$ | $3.71\pm0.02^b$ |
| Crude protein (%, DM) | $32.9\pm0.7^c$ | $37.7\pm1.2^a$ | $34.8\pm0.6^b$ |
| Lactic acid (μmol/g) | $30.0\pm0.5^b$ | $34.9\pm2.1^a$ | $31.1\pm0.8^b$ |
| Lactic acid bacteria (CFU/g FM) | $7.7\times10^8\pm3.2\times10^{7c}$ | $8.8\times10^9\pm1.7\times10^{8a}$ | $8.2\times10^9\pm1.4\times10^{8b}$ |
| Coliform bacteria (CFU/g FM) | $4.3\times10^3\pm1.6\times10^{3a}$ | $1.5\times10^2\pm4.5\times10^{1b}$ | $5.4\times10^1\pm2.00\times10^{1b}$ |
| Mould (CFU/g FM) | $4.7\times10^2\pm2.3\times10^{2a}$ | $0^b$ | $0^b$ |
| Aerobic stability (h) | $112.0\pm6.5^c$ | $141.3\pm3.8^a$ | $130.7\pm3.8^b$ |

a, b, c: means within the same line with different letters are significantly different (P < 0.05).

significantly higher (P <0.05) than that of the CT group. The lactic acid content of the T1 and T2 groups were higher than that of the CT group, but there was no significant difference between T2 and control group. In addition, the CT group had a lower number of lactic acid bacteria and a higher number of coliforms and molds (p<0.05).

## 3.3 Microbial inoculation changes the proportion of bacteria in the fermentation system

The bacterial communities of the food waste fermentation samples were determined via next-generation sequencing (NGS) and 1,147,509 raw reads of the 16S rRNA gene were obtained. After quality filtering, a total of 1,058,755 valid tags were obtained, accounting for 92.3% of the original reads, and 2267 OTUs were identified at the 97% similarity level. The Good's cover of all samples was greater than 99.5%. After normalization, the asymptote of the sparse curve of the Sobs index at the level of the operating taxon (OTU) was clear (Fig 1A), indicating that

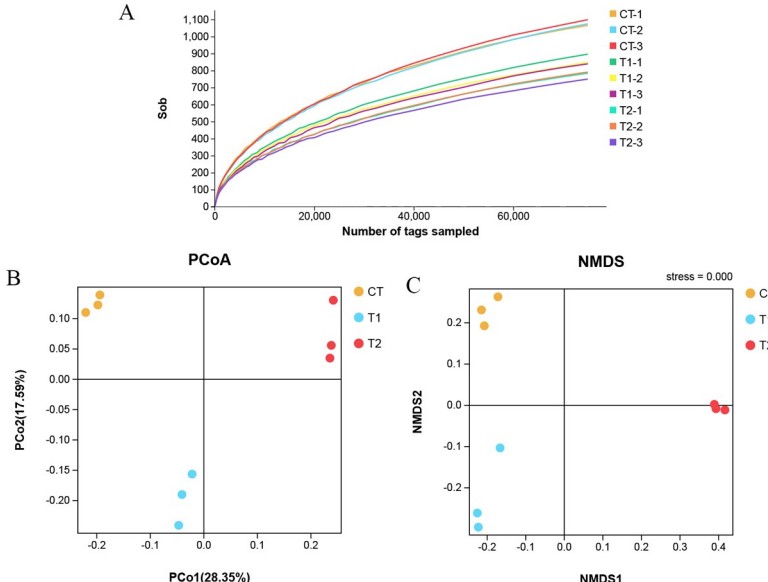

**Fig 1. Distribution of the bacterial community of food waste fermentation products.** Rarefaction curves of the 16S rRNA gene reads derived from the Sobs index at the operational taxonomic unit (OTU) level after normalization (A); Principal coordinate analysis of microflora in fermented food waste samples (B); Nonmetric multidimensional scaling of community structure (C). Each line and each point represent an individual sample. CT: uninoculated; T1: commercial bacterial inoculant; T2: laboratory-made bacterial inoculant.

**Table 3. Alpha diversity indices of the bacterial communities in food waste fermented products.**

| Item | CT | T1 | T2 | *p-value* |
|---|---|---|---|---|
| Sobs | 1151±31[a] | 899±8[b] | 834±42[b] | 0.027 |
| Chao1 | 1550±45[a] | 1337±63[ab] | 1288±57[b] | 0.039 |
| Ace | 1667±66[a] | 1393±59[b] | 1364±79[b] | 0.066 |
| Shannon | 4.09±0.02[a] | 4.12±0.02[a] | 3.23±0.01[b] | 0.067 |

a, b: means within the same line with different letters are significantly different (P < 0.05).

sampling could cover most of the bacterial diversity. Unweighted UniFrac metric-based principal coordinate analysis (PCoA) and non-metric multi-dimensional scaling (NMDS, non-metric multi-dimensional scaling, stress<0.2) can effectively reflect the differences in bacterial species between different samples. As shown in Fig 1B and 1C, the three groups of bacteria were distinctively separated at the OTU level.

For the alpha diversity analysis, a similar level of species richness existed among the T1 and T2 groups based on the Sobs, Chao1, and Ace index analyses, which indicated that there was a similar tendency of diversity and uniformity among the two groups (Table 3). Higher values for the Shannon index values indicated that the species richness and evenness in all the fermentation products (CT and T1 groups) were higher than those in the T2 group.

## 3.4 Bacterial community composition in fermentation samples

At the phylum level, a total of 20 phyla were identified (Fig 2A). Among the three treatment groups, the three most predominant phyla were Firmicutes, Cyanobacteria, and Proteobacteria. Firmicutes was the most abundant phylum in the treatment group without inoculants (CT). With the addition of bacteria, the relative abundance of Firmicutes was increased, whereas the relative abundance of Cyanobacteria and Proteobacteria was reduced.

At the genus level, a total of 244 genera were identified (Fig 2B). The 10 most predominant bacterial genera in the uninoculated group were *Lactobacillus* (82.28%), *Leuconostoc* (1.88%), *Enterococcus* (1.30%), *Lactococcus* (0.83%), *Weissella* (0.66%), *Acinetobacter* (0.34%), *Pseudomonas* (0.34%), *Lelliottia* (0.23%), *Bacillus* (0.06%), and *Acetobacter* (0.04%). Compared with the non-inoculated treatment group (CT), the commercial bacteria inoculated treatment group (T1) exhibited an increase in *Lactobacillus* (91.85%), *Acetobacter* (2.01%), and *Enterococcus* (1.63%) abundances, coupled with a decrease in *Leuconostoc* (0.20%), *Bacillus* (0.07%), *Lactococcus* (0.09%), and *Weissella* (0.07%) abundance, with a particularly strong reduction in the potentially pathogenic bacteria *Acinetobacter* (0.07%), *Pseudomonas* (0.06%), and *Lelliottia*

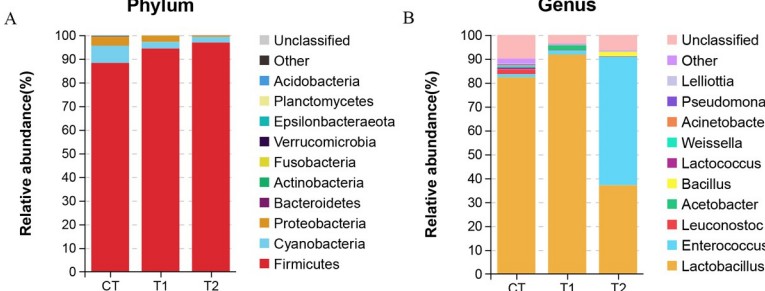

**Fig 2.** The relative abundance of bacteria community proportions at the phylum level (A) and genus level (B) across the treatment groups.

(0.03%). The most abundant bacteria in the inoculated laboratory-made bacteria treatment group (T2) were *Enterococcus* (53.81%) followed by *Lactobacillus* (37.11%) and *Bacillus* (1.85%). The high abundances of *Enterococcus* and *Bacillus* observed herein were likely due to the initial inoculation. Additionally, this group had significantly lower abundances of *Leuconostoc* (0.167%), *Acetobacter* (0.03%), *Lactococcus* (0.11%), and *Weissella* (0.06%), as well as the potentially pathogenic bacteria *Acinetobacter* (0.05%), *Pseudomonas* (0.05%), and *Lelliottia* (0.02%).

In order to evaluate the bacterial community function of the three treatment groups, PICRUSt was used to further analyze the composition of the KEGG pathways of the identified bacterial taxa. Secondary KEGG pathway analysis indicated that, compared with the CT group, T1 had an increase in the metabolism of terpenoids and polyketides (p <0.05), and a reduction in cell motility, endocrine system (p <0.05) and immune system functions (p <0.05) (Fig 3A). In contrast, the T2 group exhibited an increase in pathways associated with carbohydrate metabolism, lipid metabolism, metabolism of terpenoids and polyketides, membrane transport, infectious diseases, signal transduction, and prokaryote cellular community (p <0.05), coupled with decreases in pathways associated with the immune system (p <0.05) (Fig 3B). Compared with the T1 group, T2 exhibited significant increases in membrane transport, prokaryote cellular community, cell motility, transport and catabolism, and endocrine system (p <0.05), and decreases in digestive system (p <0.05) (Fig 3C). Therefore, the addition of bacterial agents not only changes the bacterial community structure of food waste fermentation products but also changes microorganism function.

## 3.5 Different microbial inoculants significantly affect the chemical composition of the fermentation product

In order to explore the influence of the inoculants on the fermentation products, we analyzed the metabolite composition of the three treatment products. Based on the LC-MS and GC-MS

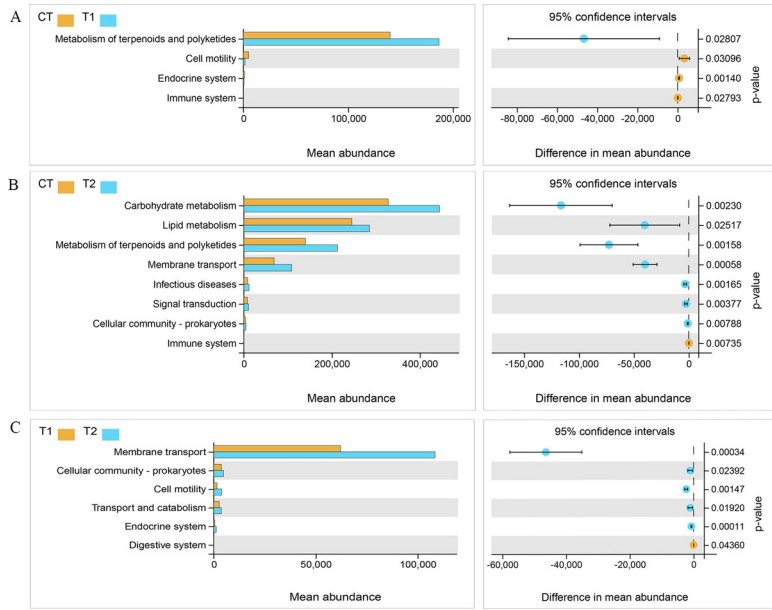

**Fig 3. KEGG metabolic pathway difference analysis.** Abundance ratios of different functional genes in the (A) T1/CT, (B) T2/CT and (C) T1/T2 groups. The middle shows the difference in functional gene abundance within a 95% confidence interval, and the right most value is the p-value (corrected).

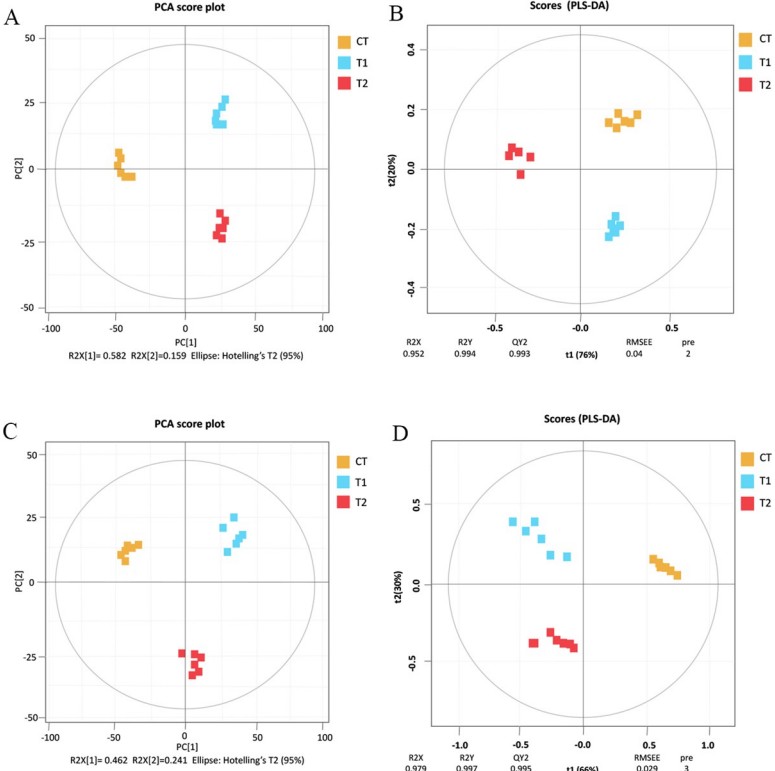

**Fig 4. Metabolome of food waste fermentation products through PCA and PLS-DA.** PCA (A) and PLS-DA (B) of the LC-MS compound metabolic spectrum to obtain the dispersion point diagram; PCA (C) and PLS-DA (D) of the GC-MS compound metabolic spectrum to obtain the dispersion point diagram.

detection results, a total of 929 substances were detected, including 853 by LC-MS and 76 by GC-MS. LC-MS identified 176 metabolites whose content was different between the T1 and CT groups. Further, 58 compounds were different between T2 and CT and 152 were different between T1 and T2. GC-MS detected 44 metabolites with different contents between T1 and CT, 46 between T2 and CT, and 42 between T1 and T2. According to PCA, the metabolites of the CT, T1, and T2 groups were clearly distinguished by PC1. The amount of variation between samples of different treatments was 58.2% (LC-MS, Fig 4A) and 46.2% (GC-MS, Fig 4C). However, multivariate analysis of PLS-DA may be more helpful to distinguish the three treatment groups. In this study, R2Ycum = 0.994 and Q2cum = 0.993 were determined for LC-MS, whereas R2Ycum = 0.997 and Q2cum = 0.995 were determined for GC-MS. This shows that the PLS-DA modeling in this study is effective and can accurately distinguish the differences in metabolites caused by different food waste treatments (Fig 4B, 4D).

**3.5.1 Differential metabolite composition identified by LC-MS promoted beneficial biological functions.** As illustrated in Fig 5 (S1 Table), after 96 h of fermentation, the samples treated with inoculants presented more amino acids (L-valine, L-tyrosine, L-phenylalanine, L-methionine, L-aspartic acid, D-proline, DL-alanine, and Beta-leucine) than the untreated samples. Compared with the control group, the inoculant-treated products exhibited a higher accumulation of the essential fatty acid linoleic acid and 9-HODE, the organic acid D-Lactin, and the nucleosides xanthine, guanine, cytosine, and adenine.

Among the identified metabolites, some substances with biological functions were also detected in the food waste fermentation products. Inoculation of commercial bacteria (T1) significantly increased the relative concentration of metabolites with antioxidant activity such as

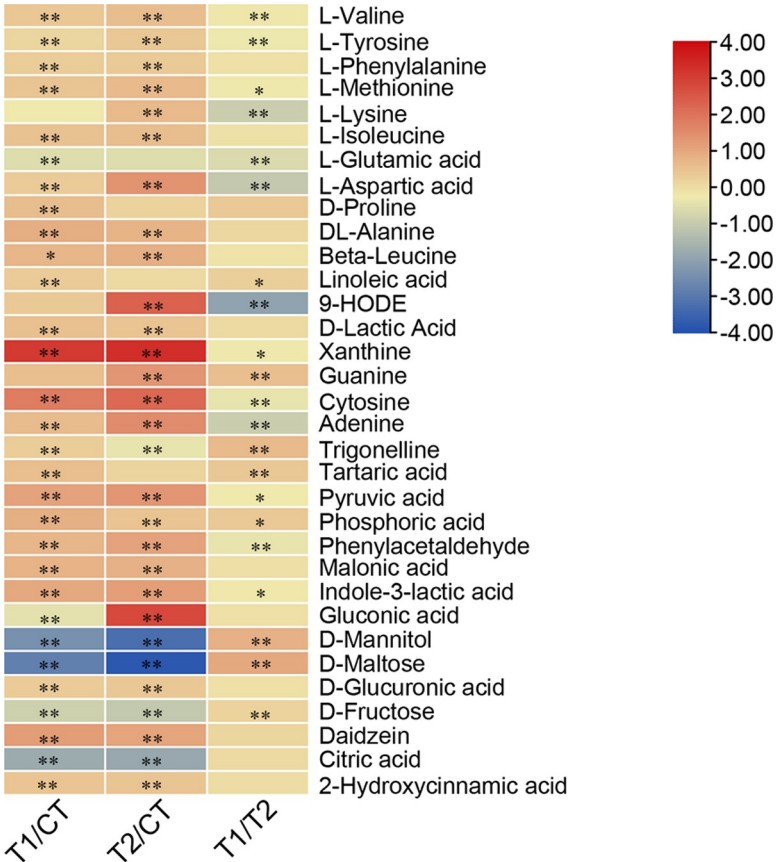

**Fig 5. Identification of significant key metabolites by LC-MS in fermented food waste sample.** The major metabolites were selected based on at least one of fold-change [$\log_2$ (T1/CT), $\log_2$ (T2/CT), $\log_2$ (T1/T2)], in addition to exhibiting a statistically significant difference. $^*0.001 < P < 0.05$; $^{**}P < 0.001$.

tartaric acid, malonic acid, and 2-hydroxycinnamic acid, the antibacterial and anti-inflammatory active compounds trigonelline and indole-3-lactic acid, and the flavoring agent phosphoric acid and daidzein, as well as pyruvic acid which improves heart function. Compared to the untreated group, the inoculated laboratory-made bacteria treatment group (T2) contained more of the antioxidant tartaric acid, malonic acid, and 2-hydroxycinnamic acid, the anti-inflammatory compound indole-3-lactic acid, the flavoring agents gluconic acid and D-glucuronic acid, and other bioactive substances such as pyruvic acid and daidzein.

**3.5.2 Microbial inoculation affects the gas components in fermentation products.** Smell is an important factor that affects the acceptability of food waste fermentation products. Therefore, this study compared the volatile gas components of food waste-derived fermentation products through GC-MS. As shown in Fig 6 (S2 Table), the identified differential compounds included pleasant aroma compounds, pungent odor compounds, and some odorless compounds.

Group T1 and T2 exhibited high abundances of phenylethyl alcohol, acetic acid, and 1-butanol, all of which impart pleasant fragrances. Group CT had high abundances of β-myrcene, linalool, geraniol, eucalyptol, anethole, 1-hexanol, and other fragrance molecules. It is worth noting that phenol concentrations increased in both inoculant treatments, especially in T1. In contrast, group CT exhibited the highest benzene concentrations compared to T1 and T2.

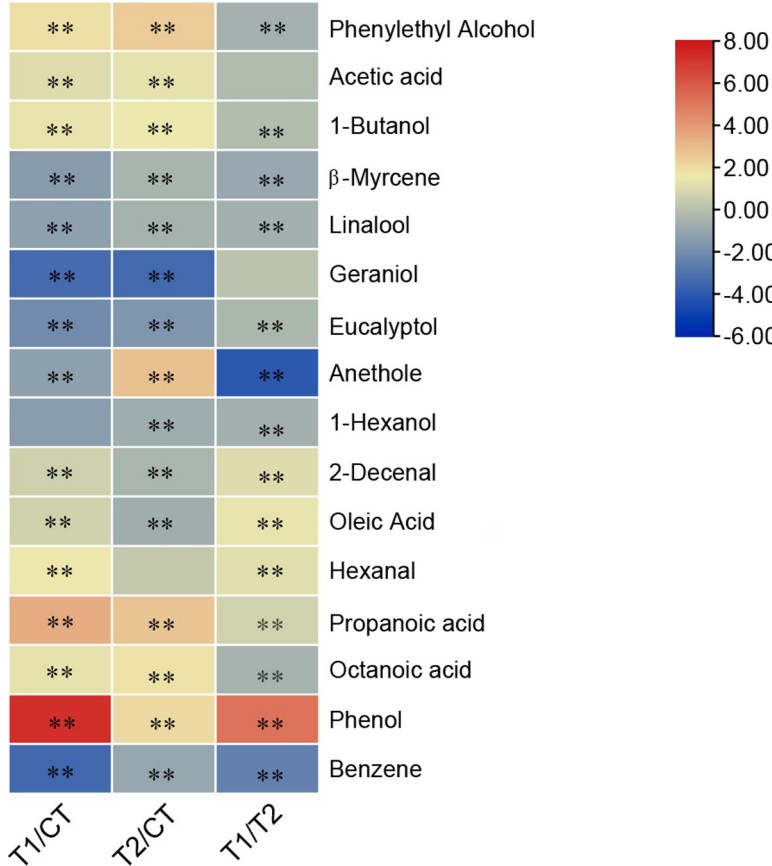

**Fig 6. Identification of significant key metabolites by GC-MS in fermented food waste sample.** The major metabolites were selected based on fold-change [log$_2$ (T1/CT), log$_2$ (T2/CT), log$_2$ (T1/T2)], in addition to exhibiting a statistically significant difference. $^*0.001 < P < 0.05$; $^{**}P < 0.001$.

## 3.6 Correlation between the relative abundance of bacteria in fermentation products and metabolites

As illustrated in Fig 7, the dominant genus, *Lactobacillus*, was only negatively related to two kinds of amino acids (L-lysine and L-aspartic acid), 9-HODE, and Guanine. This indicated that the difference in metabolites between different treatments may not be caused by the dominant species. *Enterococcus* was positively correlated with L-lysine, L-aspartic acid, and 9-HODE, indicating that this genus improves the nutritional value of food waste. However, *Enterococcus* was also positively correlated with the nucleosides guanine and adenine. *Leuconostoc* was positively correlated with four kinds of functional compounds (D-mannitol, D-maltose, D-fructose, and citric acid). These correlations suggested that *Leuconostoc* would be a promising candidate species to enhance the quality of fermented products, although this genus was negatively correlated with two kinds of amino acids (L-phenylalanine, DL-alanine), an organic acid (D-lactic acid), two kinds of nucleotides (xanthine and cytosine), and six kinds of functional compounds (pyruvic acid, phenylacetaldehyde, malonic acid, indole-3-lactic acid, daidzein, and 2-hydroxycinnamic acid). *Acetobacter* was negatively correlated with L-glutamic acid. *Bacillus* was positively correlated with L-aspartic acid, 9-HODE, guanine, and adenine. *Lactococcus*, *Weissella*, *Acinetobacter*, *Pseudomonas*, *Lelliottia*, and others exhibited a positive correlation with four kinds of functional compounds (D-mannitol, D-maltose, D-fructose,

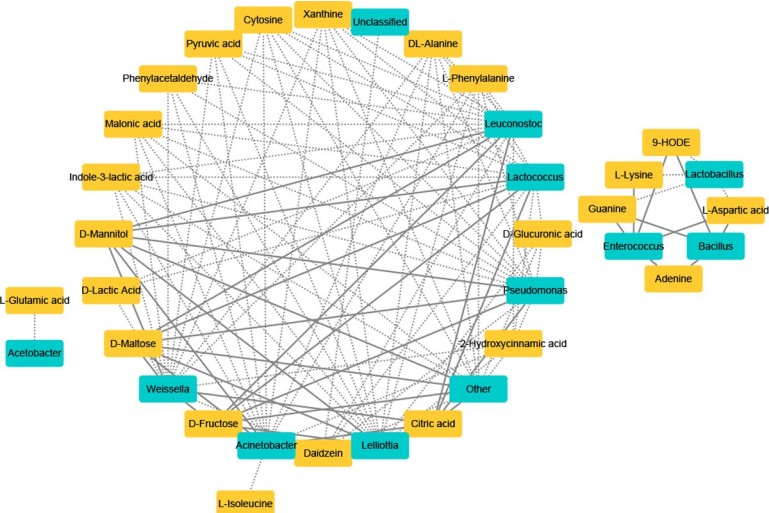

**Fig 7. Spearman correlations between metabolites by LC-MS analysis and main bacteria species.** Differentially expressed metabolites during fermentation were screened by PLS-DA; Positive correlations are indicated with solid lines (R > 0.90) and negative correlations are shown by dashed lines (R < -0.90).

and citric acid); at the same time, they were negatively correlated with two kinds of amino acids (L-phenylalanine and DL-alanine), xanthine, and four kinds of functional compounds (malonic acid, indole-3-lactic acid, daidzein, and 2-hydroxycinnamic acid). Additionally, *Lactococcus* and *Weissella* were negatively correlated with D-lactic acid. *Lactococcus*, *Pseudomonas*, *Lelliottia*, and others were negatively correlated with two kinds of functional compounds (D-glucuronic acid and daidzein). Except for "others," other dominant genera were negatively correlated with pyruvic acid. *Acinetobacter* was negatively correlated with L-isoleucine.

We also analyzed the correlation between bacterial flora and changes in volatile gas composition. As shown in Fig 8, several correlations were identified between bacterial taxa and compounds identified via GC-MS. *Lactobacillus* was negatively correlated with the aroma substance Anethole, whereas *Enterococcus* was positively correlated with this compound.

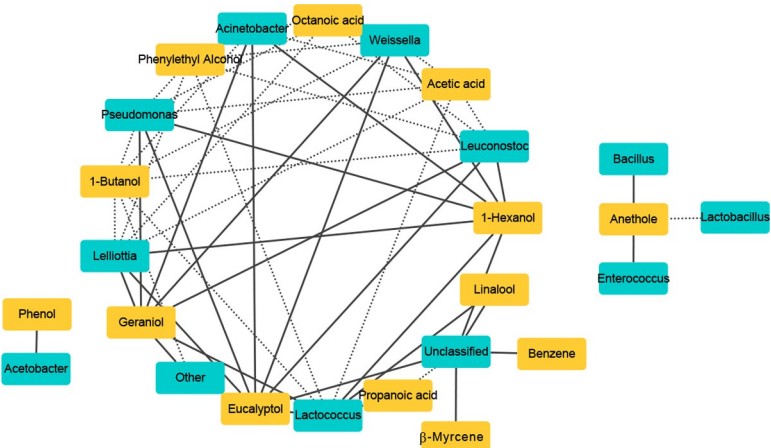

**Fig 8. Spearman correlations between metabolites by GC-MS analysis and main bacteria species.** Differentially expressed metabolites during fermentation were screened by PLS-DA; Positive correlations are indicated with solid lines (R > 0.90) and negative correlations are shown by dashed lines (R < -0.90).

*Leuconostoc* was positively correlated with three aroma substances (1-Hexanol, Eucalyptol, Geraniol) and was negatively correlated with three aroma substances (Acetic acid, 1-Butanol, Phenylethyl alcohol) and Octanoic acid, which produces a rancid smell. *Acetobacter* was positively correlated with Phenol. *Bacillus* was positively correlated with Anethole. *Lactococcus*, *Weissella*, *Acinetobacter*, *Pseudomonas*, and *Lelliottia* were positively related to three kinds of aroma substances (1-Hexanol, Eucalyptol, Geraniol) and were also negatively related to three kinds of aroma substances (1-Butanol, Acetic acid, Phenylethyl alcohol), in addition to one negatively related odor substance (Octanoic acid). Additionally, *Lactococcus* was positively correlated with the aroma substance Linalool and negatively correlated with the odor substance Propanoic acid.

## 4. Discussion

Food waste fermentation is a complex process dominated by microbial community. Bacterial community analysis and metabolomic analysis of fermentation systems can provide important insights into animal health and welfare from the perspective of nutritional value, as well as important information that enables the screening of microorganisms that regulate food waste fermentation [14, 29]. This study was the first to combine bacterial community and metabolomic analyses to elucidate the bacterial community and metabolome characteristics of fermentation products derived from commercial and laboratory-made bacteria inoculated with food waste from a centralized treatment facility. This study also revealed that the dominant microorganisms have different roles in the fermentation of food waste.

### 4.1 Characteristics of food waste fermentation products

The heat-treated food waste samples in this study were similar to those described in previous studies. Concretely, the samples were slippery in appearance, exhibited a brown or black coloration, and had a fat aroma without a foul smell [10]. After 96 h of treatment, all three treatment groups (no inoculation, inoculation with a commercial inoculant, and inoculation with a laboratory-made inoculant) showed a greater improvement in color and odor compared to the heat-treated products. In terms of color, the samples developed a yellowish coloration, which contrasted with the black color of the raw material. The laboratory-made inoculant treatment group exhibited the yellowest coloration, followed by the commercial bacteria treatment group. We speculated that this was caused by the addition of excipients and the effect of internal strains. Moreover, the crude protein content increased in the treatment groups (T1 and T2) with the addition of the bacterium, indicating an increase in nutritional value. These samples also contained more lactic acid bacteria, higher lactic acid contents, lower pH, lower mold content, and higher aerobic stability ($p < 0.05$). This result is similar to that of Du et al. [14] for the production of fermented feed from cabbage waste, indicating that exogenous probiotic inoculation is a promising strategy to enhance the bioconversion of food waste to animal feed.

### 4.2 Effects of inoculants on bacterial microbiota in food waste fermentation products

At the genus level, the genus *Lactobacillus* includes most food-fermenting lactic acid bacteria, which can metabolize sugar into lactic acid and are used as starters in the industrial production of fermented foods and animal feed [30]. Further, *Enterococcus* is often found in fermented feed. For example, Jin et al. [15] demonstrated that *Enterococcus* was the most abundant bacterial genus after fermentation instead of the added probiotic *Lactobacillus* or *Bacillus*. *Lactococcus* is generally considered safe and is therefore commonly used in the dairy industry [31]. *Leuconostoc* spp. belong to the LAB functional group and can actively participate in the

fermentation process, especially in the production of sauerkraut and kimchi. Importantly, these microorganisms are considered safe by the US Food and Drug Administration (FDA) [32]. *Weissella* is commonly detected in animals but is also found in vegetables and various fermented foods such as European sourdough and traditional fermented foods in Asia and Africa. Certain *Weissella* strains have also garnered attention as potential probiotics [33]. The members of the genus *Bacillus* can effectively exert antibacterial activity in the gastrointestinal tract and are therefore considered probiotics [34]. *Acetobacter* is used in industrial vinegar production due to its strong ability to oxidize ethanol into acetic acid coupled with its strong acetic acid resistance [35]. However, *Acinetobacter*, *Pseudomonas*, and *Lelliottia* are considered potentially harmful. *Acinetobacter* is strictly aerobic and it is commonly linked to infections in frail patients in hospitals [36]. Some members of the *Pseudomonas* genus act as opportunistic pathogens, causing a variety of infectious diseases in animals and humans, in addition to their role as plant pathogens and specific spoilage microorganisms [37]. The genus *Lelliottia* is a new genus whose members were previously classified as *Enterobacter*. Currently, only two species have been reported, and both are thought to cause diseases [38]. Compared with the non-inoculated treatment group (CT), the commercial bacteria inoculated treatment group (T1) and laboratory-made bacteria inoculated treatment group (T2) exhibited a strong reduction in the abundance of the potentially pathogenic bacteria *Acinetobacter*, *Pseudomonas*, and *Lelliottia*. This indicates that the addition of commercial or laboratory-made bacteria can inhibit the growth of undesirable microorganisms. Therefore, inoculation can improve the nutritional value and fermentation quality of food waste, thus benefiting animal health and welfare.

## 4.3 Effects of inoculants on the metabolomic profiles of food waste fermentation products

Our results indicated that the samples treated with the inoculants exhibited an increased accumulation of some free amino acids, which is consistent with previous studies on food waste fermentation [13]. In addition to amino acids, our metabolomic analyses identified the presence of several essential fatty acids, organic acids, and a variety of metabolites with specific biological functions in the food waste fermentation products, including phosphoric acid, citric acid, and maltose, which are currently used as feed additives, as well as antibacterial substances, flavoring agents, and other additives that contribute to animal health and welfare. Further, our study identified high concentrations of metabolites with biological functions in the groups with added bacteria, including compounds with antibacterial, antioxidant, and anti-inflammatory activities such as tartaric acid, indole-3-lactic acid [39, 40], and 2-hydroxycinnamic acid [41], as well as phosphoric acid, phenylacetaldehyde, malonic acid, which possess taste modifying properties, pyruvic acid, which stimulates metabolism and increases cardiac function, D-gluconic acid, which has detoxifying properties, and daidzein, which reduces the risk of certain hormone-related cancers and heart disease [42–44]. In contrast, trigonelline, gluconic acid, D-mannitol, D-maltose, D-fructose, and citric acid were more abundant in the uninoculated treatment group than in the T1 or T2 groups. These biologically active metabolites may have been produced by bacteria originally present in the CT group.

The fermentation products of the different treatments had different compositions of odor molecules. Both the T1 and T2 groups exhibited high levels of phenyl alcohol, acetic acid, and 1-butanol, all of which produced pleasant aromas. The CT group was rich in aromatic molecules such as β-laurolene, linalool, geraniol, eucalyptol, anisyl alcohol, and 1-hexanol. It is also worth noting that phenol concentrations increased in both inoculation treatments, especially in the T1 group. Phenol is a harmful substance that severely irritates the eyes, skin, and respiratory tract, in addition to potentially causing harmful effects on the central nervous system,

reproductive system, heart, kidney, and during embryonic development [45, 46]. In contrast, the CT group exhibited the highest benzene concentrations compared to T1 and T2. Benzene has a sweet and aromatic odor but has been linked to several acute and long-term adverse health effects and diseases including acute myeloid leukemia and cancer [47, 48]. Therefore, phenol and benzene contents in fermentation products must be strictly controlled.

### 4.4 Correlation between the bacterial populations and metabolites in the food waste fermentation products

According to our results, *Enterococcus* was also positively correlated with the nucleosides guanine and adenine. Excessive intake of purines may increase the risk of hyperuricemia and gout, and therefore the levels of purines in animal-derived foods are becoming an increasing concern [48]. *Leuconostoc* was positively correlated with four kinds of functional compounds. These correlations suggested that *Leuconostoc* would be a promising candidate species to enhance the quality of fermented products. *Bacillus* was positively correlated with an amino acids, a kind of fatty acid and a kind of nucleoside. *Lactococcus*, and *Weissella* exhibited a positive correlation with four kinds of functional compounds. These correlations suggest that *Leuconostoc*, *Lactococcus*, and *Weissella* would be promising candidates to improve the quality of fermentation products.

We also found that *Enterococcus* was positively correlated with the aroma substance anethole. *Leuconostoc* was positively correlated with three aroma substances (1-hexanol, eucalyptol, and geraniol) and was negatively correlated with octanoic acid, a rancid smell compound. Similarly, *Bacillus* was positively correlated with anethole. *Lactococcus* and *Weissella* were positively correlated with three kinds of aroma substances (1-hexanol, eucalyptol, and geraniol) and were also negatively correlated with octanoic acid, a substance that confers a pungent odor. Additionally, *Lactococcus* was positively correlated with the aroma substance linalool and negatively correlated with the odorous substance propanoic acid. However, *Acetobacter* was positively correlated with phenol. Therefore, *Enterococcus*, *Leuconostoc*, *Bacillus*, *Lactococcus*, and *Weissella* have been suggested as alternative genera to improve food waste odor. The results of the metabolic profile (LC-MS and GC-MS) and microbiome correlation revealed that *Leuconostoc*, *Lactococcus* and *Weissella* would be promising candidates for improving the quality of fermentation products.

However, given that correlation does not equal causation, further statistics and correlation parameters are required to confirm the aforementioned speculations. Additionally, it should be noted that only three samples were analyzed per group for the microbiome assessments and six samples per group for the metabolome assessments. Therefore, the conclusions that might be drawn from such a small sample size are limited.

## 5. Conclusion

Microbial inoculation increased the crude protein content of food waste while reducing the pH value, increasing lactic acid production, and enhancing aerobic stability. Moreover, microbial inoculation affected the diversity and abundance of microbial communities and reduced the abundance of potentially pathogenic bacteria. This process also changed the metabolite profile, producing many beneficial metabolites and volatile odors, but also increased the abundance of undesirable odors and some harmful substances. It is hypothesized that *Leuconostoc*, *Lactococcus*, and *Weissella* would be promising candidates to improve the quality of fermentation products. Taken together, our findings may improve our overall understanding of food waste fermentation and contribute to the development of novel inoculum formulations.

## Supporting information

**S1 Fig. Schematic diagram of the experimental setup.**
(TIF)

**S1 Table. Identification of significant key metabolites by LC-MS in fermented food waste sample.**
(DOCX)

**S2 Table. Identification of significant key metabolites by GC-MS in fermented food waste sample.**
(DOCX)

## Acknowledgments

We are grateful for the sequencing platform of Gene Denovo Biotechnology Co., Ltd. (Guangzhou, China). Thanks to mjeditor (www.mjeditor.com) for its linguistic assistance during the manuscript writing.

## Author Contributions

**Conceptualization:** Hongmei Li, Lujun Yu, Yongxue Sun.

**Data curation:** Hongmei Li, Zongyu Miao, Yuanzheng Wei, Jin Zeng.

**Formal analysis:** Xiaoyang Lin, Lujun Yu.

**Funding acquisition:** Ren Huang.

**Investigation:** Hongmei Li, Xiaoyang Lin, Lujun Yu, Zongyu Miao, Yuanzheng Wei, Jin Zeng.

**Resources:** Jianjun Li, Qi Zhang, Ren Huang.

**Supervision:** Jianjun Li, Yongxue Sun, Ren Huang.

**Writing – original draft:** Hongmei Li.

**Writing – review & editing:** Ren Huang.

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
