## [Decision Letter · Decision Letter 0]

1 Dec 2021

PONE-D-21-35732Comprehensive
characterization of the bacterial community structure and metabolite composition of
food waste fermentation products via microbiome and metabolome
analysesPLOS ONE

Dear Dr. Li,

Thank you for submitting your manuscript to PLOS ONE. After careful consideration, we
feel that it has merit but does not fully meet PLOS ONE’s publication criteria as it
currently stands. Therefore, we invite you to submit a revised version of the
manuscript that addresses the points raised during the review process.

Please submit your revised manuscript by Jan 15 2022 11:59PM. If you will need more
time than this to complete your revisions, please reply to this message or contact
the journal office at plosone@plos.org. When
you're ready to submit your revision, log on to https://www.editorialmanager.com/pone/ and select the 'Submissions
Needing Revision' folder to locate your manuscript file.

Please include the following items when submitting your revised
manuscript:A rebuttal letter that responds to each point raised by the academic
editor and reviewer(s). You should upload this letter as a separate file
labeled 'Response to Reviewers'.A marked-up copy of your manuscript that highlights changes made to the
original version. You should upload this as a separate file labeled
'Revised Manuscript with Track Changes'.An unmarked version of your revised paper without tracked changes. You
should upload this as a separate file labeled 'Manuscript'.

If you would like to make changes to your financial disclosure, please include your
updated statement in your cover letter. Guidelines for resubmitting your figure
files are available below the reviewer comments at the end of this letter.

We look forward to receiving your revised manuscript.

Kind regards,

Guanglei Qiu

Academic Editor

PLOS ONE

Journal Requirements:

" ext-link-type="uri"
xlink:type="simple">https://journals.plos.org/plosone/s/file?id=ba62/PLOSOne_formatting_sample_title_authors_affiliations.pdf"

Additional Editor Comments:

As you may see from the reviewers' comments, generally they are positive about the
overall merits of your work. But still there are rooms for further improvements.
Please revise the MS carefully based on the reviewers' comments and suggestions. A
final decision will be made largely based on the reviewers' re-evaluation of your MS
after revision.

Reviewers' comments:

Reviewer's Responses to Questions

**Comments to the Author**

1. Is the manuscript technically sound, and do the data support the conclusions?

Reviewer #1: Yes

Reviewer #2: Yes

2. Has the statistical analysis been performed
appropriately and rigorously? 

Reviewer #1: Yes

Reviewer #2: Yes

3. Have the authors made all data underlying the
findings in their manuscript fully available?

Reviewer #1: Yes

Reviewer #2: Yes

4. Is the manuscript presented in an intelligible
fashion and written in standard English?

Reviewer #1: Yes

Reviewer #2: Yes

5. Review Comments to the Author

Reviewer #1: This study intended to comprehensive characterize the bacterial
community and metabolite composition of food waste fermentation products by
microbiome and metabolome analyses. Overall the article presents enough data for a
paper. The following suggestions should be helpful and considered for the
significant improvement of the manuscript as well as their future study.

1. In the Abstract, the commercial bacterial agent and 2% of a laboratory-made
bacterial agent can be simply introduced for more details.

2. In the Abstract, the authors wrote “Microbial inoculation also affected the
diversity and abundance of microbial communities”. However, how did the microbial
inoculation affect the microbial abundance indices?

3. The Introduction part should be improved. For example, the detailed amount of the
production of the food waste in China can be mentioned.

4. The authors should critically check the correctness of some descriptions. For
example, the authors wrote that “There are no studies on the microbial communities
and metabolic profiles of fermented food wastes of various sources and complex
compositions processed in centralized treatment facilities” the microbial community
should have been widely reported in my humble view since the Miseq sequencing has
been widely used since 2011, while the food fermentation was also a common topic.
So, it is too absolute to say that the microbial communities have not been
reported.

5. Why did the microbial inoculation reduce the alpha diversity indexes?

6. The reasons for the production of undesirable odors and some harmful substances
can be explained based on the results.

7. The authors applied 0.18% of a commercial bacterial agent and 2% of a
laboratory-made bacterial agent. Since 2% was ten times more than 0.18%, why did the
authors choose a relatively high inoculation rate?

8. The experimental setup seemed to be not described clearly, which can be presented
in the SI.

Reviewer #2: Overview recommendation:

In this study, the microbial community and metabolite profile were investigated by
16S rRNA gene sequencing, GC-MS and LC-MS during the solid food waste fermented
process. Meanwhile, the bacterial agent was used to improve the fermented products.
The study is meaningful, however the description and analysis in the section of
Results is not enough. And the cacography existed in the manuscript. Therefore, the
paper need to be revised before publishing.

General recommendation:

1. Page 8, the section of Abstract: Suggest that the data related results should be
added into the Abstract, rather than description without data.

2. Page 8, the section of Abstract: the name of genus, such as Leuconostoc,
Lactococcus and Weissella, should be wrote in italic.

3. Page 9, the section of Introduction, the second paragraph: the sentence of “Chen
collected food wasted…animal feeds was presented” is confuse, please revised it.

4. Page 11, the section of 2.1, the 5th line: the initial of “Samples” should be
lower-case. The 8th line: “8% corn meal and 5% soybean meal powder” is mass ratio or
volume ratio?

5. The vertical spacing in Table 1 and Table 2 should be unification.

6. The section of 3.6, the second paragraph, 6th-7th line: the initial of “1-hexanol”
and “octanoic acid” should be the capital form. The 9th line: “Anethole” should not
be italic, please revised it.

7. The food waste samples was inoculated with 0.18% commercial inoculum 1 (T1) and 2%
laboratory-made inoculum 2 (T2). Whether there is comparability? Because the
additive amounts of bacterial agent were different.

8. How much is cost of additional bacterial agent when treatment of solid food waste
fermented products? Is the price of bacterial agent acceptable?

9. Is the performance of bacterial agent durable?

6. PLOS authors have the option to publish the peer
review history of their article (what does this mean?). If published, this will
include your full peer review and any attached files.

If you choose “no”, your identity will remain anonymous but your review may still be
made public.

**Do you want your identity to be public for this peer review?** For
information about this choice, including consent withdrawal, please see our
Privacy Policy.

Reviewer #1: No

Reviewer #2: No

---

## [Author Response · Author response to Decision Letter 0]

11 Jan 2022

Reviewer #1: This study intended to comprehensive characterize the bacterial
community and metabolite composition of food waste fermentation products by
microbiome and metabolome analyses. Overall the article presents enough data for a
paper. The following suggestions should be helpful and considered for the
significant improvement of the manuscript as well as their future study.

1.In the Abstract, the commercial bacterial agent and 2% of a laboratory-made
bacterial agent can be simply introduced for more details.

Answer: Thank you for your suggestions. The required details have been included in
the abstract as follows: “inoculated with or without 0.18% of a commercial bacterial
agent consisting of multiple unknown strains and 2% of a laboratory-made bacterial
agent consisting of Enterococcus faecalis, Bacillus subtilis, and Candida utilis.” 

2.In the Abstract, the authors wrote “Microbial inoculation also affected the
diversity and abundance of microbial communities”. However, how did the microbial
inoculation affect the microbial abundance indices?

Answer: Thank you for your insightful observations. Microbial inoculation affected
the community richness, and the Sobs and Ace indices were lower (P 0.05) in the
groups inoculated with the commercial bacterial agent and laboratory-made bacterial
agent than in the non-inoculated treatment group. Microbial inoculation also
affected the community diversity, and the Shannon index of the group inoculated with
the laboratory-made bacterial agent was lower (P 0.05) than that of the group
inoculated with the commercial bacterial agent and the uninoculated group. 

3.The Introduction part should be improved. For example, the detailed amount of the
production of the food waste in China can be mentioned.

Answer: Thank you for your suggestions. Liu et al. reported that food stalls,
restaurants, and canteens in China generate approximately 45 million tons of food
waste per year. This information has been included in the Introduction.

4.The authors should critically check the correctness of some descriptions. For
example, the authors wrote that “There are no studies on the microbial communities
and metabolic profiles of fermented food wastes of various sources and complex
compositions processed in centralized treatment facilities” the microbial community
should have been widely reported in my humble view since the Miseq sequencing has
been widely used since 2011, while the food fermentation was also a common topic.
So, it is too absolute to say that the microbial communities have not been
reported.

Answer: Thank you for your suggestions. We agree that the presentation of that
paragraph in the article was confusing. The point we were trying to make is that no
one has yet analyzed the fermentation products of food waste treated in centralized
treatment centers using 16SrRNA and metabolomic methods. We have improved the
presentation of that section.

5.Why did the microbial inoculation reduce the alpha diversity indexes?

Answer: The commercial bacterial agent or laboratory-made bacterial agent added to
the food waste medium are more likely to form the dominant flora. They will secrete
some metabolites to prevent the growth and reproduction of others. Therefore,
compared with the control group, the diversity of the inoculated treatment groups
was reduced.

6.The reasons for the production of undesirable odors and some harmful substances can
be explained based on the results.

Answer: Thank you for your question. Microbial inoculation reduced the pH value of
the fermentation product, making the lipids with the benzene ring easier to
hydrolyze, resulting in higher phenol production rates under acidic conditions.
Additionally, Acetobacter was positively correlated with phenol and Unclassified was
positively correlated with benzene. Hexanal, propionic acid, and octanoic acid
(compounds with undesirable odors) may be produced mainly by the metabolism of some
bacteria, such as Acetobacter, Enterococcus, and Bacillus.

7.The authors applied 0.18% of a commercial bacterial agent and 2% of a
laboratory-made bacterial agent. Since 2% was ten times more than 0.18%, why did the
authors choose a relatively high inoculation rate?

Answer: Thank you for your question. The commercial bacterial agent purchased is a
freeze-dried product and the amount added for fermentation is a mass ratio. The
addition of the commercial bacterial agent resulted in a larger increase in
production costs. In order to save costs and facilitate production, the
laboratory-made bacterial agent used in this experiment was collected by
centrifugation and was not lyophilized.

8. The experimental setup seemed to be not described clearly, which can be presented
in the SI.

Answer: Thank you for your suggestions. The relevant experimental setup has been
described in detail in Figure S1.

Reviewer #2: Overview recommendation:

In this study, the microbial community and metabolite profile were investigated by
16S rRNA gene sequencing, GC-MS and LC-MS during the solid food waste fermented
process. Meanwhile, the bacterial agent was used to improve the fermented products.
The study is meaningful, however the description and analysis in the section of
Results is not enough. And the cacography existed in the manuscript. Therefore, the
paper need to be revised before publishing.

General recommendation:

1. Page 8, the section of Abstract: Suggest that the data related results should be
added into the Abstract, rather than description without data.

Answer: Thank you for your suggestion. The related results have been included in the
abstract.

2.Page 8, the section of Abstract: the name of genus, such as Leuconostoc,
Lactococcus and Weissella, should be wrote in italic.

Answer: Thank you for your suggestion. We have changed these words to italics in the
Abstract, as well as throughout our entire manuscript.

3.Page 9, the section of Introduction, the second paragraph: the sentence of “Chen
collected food wasted…animal feeds was presented” is confuse, please revised it.

Answer: Thank you for your advice. I changed this sentence into “Three typical
treatment processes (i.e., heat treatment, fermentation, and coupled hydrothermal
treatment and fermentation) are usually used in centralized food waste treatment
centers. Food waste processed using either of the aforementioned procedures is
considered to have some nutritional value and meets relevant microbiological and
chemical contaminant standards, making food waste a promising alternative to be used
in animal diets.” 

4.Page 11, the section of 2.1, the 5th line: the initial of “Samples” should be
lower-case. The 8th line: “8% corn meal and 5% soybean meal powder” is mass ratio or
volume ratio?

Answer: Thank you for your suggestions. The word "Samples" is now in lower case.
Regarding the composition ratio of food waste and auxiliary materials, 87% was food
waste heat treatment material, 8% was soybean meal, and 5% was cornmeal. These
proportions are reported on a mass ratio basis. 

5.The vertical spacing in Table 1 and Table 2 should be unification.

Answer: Thank you for your suggestions. The vertical spacing of Tables 1 and 2 is now
equal.

6.The section of 3.6, the second paragraph, 6th-7th line: the initial of “1-hexanol”
and “octanoic acid” should be the capital form. The 9th line: “Anethole” should not
be italic, please revised it.

Answer: Thank you for your suggestions. All corrections have been implemented in the
revised version of the manuscript.

7.The food waste samples was inoculated with 0.18% commercial inoculum 1 (T1) and 2%
laboratory-made inoculum 2 (T2). Whether there is comparability? Because the
additive amounts of bacterial agent were different.

Answer: Thank you for your question. The purchased commercial bacterial agent is
available in lyophilized form and is added at a mass ratio of 0.18% of the food
waste fermentation medium. The laboratory-made bacterial agent is obtained by
centrifugal collection at 2% of the food waste fermentation medium. Many previous
articles refer to the addition of centrifugal bacteria at 1–5% of the fermentation
composition. In a preliminary study, we compared the fermentation effectiveness and
cost of several different additions and decided to use a 2% inoculum.

8.How much is cost of additional bacterial agent when treatment of solid food waste
fermented products? Is the price of bacterial agent acceptable?

Answer: Thank you for your questions. In the process of converting kitchen waste to
feed, the cost of the bacterium is an important consideration. Adding 0.18% of
bacteriological agent increases the treatment cost by an additional 120 RMB per ton.
For solid waste treatment, the cost is too high and harder for companies to bear. In
contrast, the laboratory-made bacterium formulation could be prepared by the factory
itself. The liquid medium can be used for the homogenization of kitchen waste
without resulting in additional pollution and waste. The increased cost is less than
1/5 of the commercial bacteriological agent.

9.Is the performance of bacterial agent durable?

Answer: We conducted colony counts of Lactobacillus, Bacillus, and yeasts in
fermentation products obtained using the laboratory-made bacterial inoculant. The
colony counts of Lactobacillus and Bacillus were within the same order of magnitude
at 4, 14, and 21 days after fermentation. However, the number of yeast colonies
after 14 and 21 days was significantly lower than that of colonies at 4 days
post-fermentation.

to Reviewers.docx
---

## [Decision Letter · Decision Letter 1]

7 Feb 2022

Comprehensive characterization of the bacterial community structure and metabolite
composition of food waste fermentation products via microbiome and metabolome
analyses

PONE-D-21-35732R1

Dear Dr. Li,

We’re pleased to inform you that your manuscript has been judged scientifically
suitable for publication and will be formally accepted for publication once it meets
all outstanding technical requirements.

Kind regards,

Guanglei Qiu

Academic Editor

PLOS ONE

Additional Editor Comments (optional):

Reviewers' comments:

Reviewer's Responses to Questions

**Comments to the Author**

1. If the authors have adequately addressed your comments raised in a previous round
of review and you feel that this manuscript is now acceptable for publication, you
may indicate that here to bypass the “Comments to the Author” section, enter your
conflict of interest statement in the “Confidential to Editor” section, and submit
your "Accept" recommendation.

Reviewer #1: (No Response)

Reviewer #2: All comments have been addressed

2. Is the manuscript technically sound, and do the data
support the conclusions?

Reviewer #1: (No Response)

Reviewer #2: Yes

3. Has the statistical analysis been performed
appropriately and rigorously? 

Reviewer #1: (No Response)

Reviewer #2: Yes

4. Have the authors made all data underlying the
findings in their manuscript fully available?

Reviewer #1: (No Response)

Reviewer #2: Yes

5. Is the manuscript presented in an intelligible
fashion and written in standard English?

Reviewer #1: (No Response)

Reviewer #2: Yes

6. Review Comments to the Author

Reviewer #1: The reviewers have revised the manuscript based on the suggestions and
it is much more publishable now.

Reviewer #2: (No Response)

7. PLOS authors have the option to publish the peer
review history of their article (what does this mean?). If published, this will
include your full peer review and any attached files.

If you choose “no”, your identity will remain anonymous but your review may still be
made public.

**Do you want your identity to be public for this peer review?** For
information about this choice, including consent withdrawal, please see our
Privacy Policy.

Reviewer #1: No

Reviewer #2: **Yes: **Haibo Li

---

## [Editor Report · Acceptance letter]

7 Mar 2022

PONE-D-21-35732R1 

Comprehensive characterization of the bacterial community structure and metabolite
composition of food waste fermentation products via microbiome and metabolome
analyses 

Dear Dr. Li:

I'm pleased to inform you that your manuscript has been deemed suitable for
publication in PLOS ONE. Congratulations! Your manuscript is now with our production
department. 

Kind regards, 

on behalf of

Dr. Guanglei Qiu 

Academic Editor

PLOS ONE